# Postoperative Pain Is Driven by Preoperative Pain, Not by Endometriosis

**DOI:** 10.3390/jcm10204727

**Published:** 2021-10-15

**Authors:** Panagiotis Kanellos, Konstantinos Nirgianakis, Franziska Siegenthaler, Christian Vetter, Michael D. Mueller, Sara Imboden

**Affiliations:** 1Department of Gynecology and Obstetrics, Bern University Hospital, University of Bern, 3010 Bern, Switzerland; panoskanellos@hotmail.com (P.K.); Konstantinos.Nirgianakis@insel.ch (K.N.); franziska.siegenthaler@insel.ch (F.S.); chefarztsekretariat.gynaekologie@insel.ch (M.D.M.); 2Department of Anesthesiology and Pain Medicine, Bern University Hospital, University of Bern, 3010 Bern, Switzerland; christian.vetter@insel.ch

**Keywords:** chronic pelvic pain, endometriosis, laparoscopic hysterectomy, postoperative pain

## Abstract

(1) Background: The aim of this study was to evaluate the impact of endometriosis on postoperative pain following laparoscopic hysterectomy; (2) Methods: A total of 214 women who underwent a laparoscopic hysterectomy between January 2013 and October 2017 were divided into four subgroups as follows: (1) endometriosis with chronic pain before the surgery (*n* = 57); (2) pain-free endometriosis (*n* = 50); (3) pain before the surgery without endometriosis (*n* = 40); (4) absence of both preoperative pain and endometriosis (*n* = 67). Postoperative pain was compared by using Visual Analog Scale (VAS) scores and by tracking the use of painkillers during the day of surgery and the first two postoperative days; (3) Results: Women with chronic pain before the surgery reported higher VAS scores during the first postoperative days, while the use of analgesics was similar across the groups. There was no difference in the postoperative pain when comparing endometriosis patients to non-endometriosis patients; (4) Conclusions: Women with chronic pelvic pain demonstrated increased postoperative pain after laparoscopic hysterectomy, which was independent of the presence or severity of endometriosis. The increased VAS scores did not, however, translate into equally greater use of painkillers, possibly due to the standardised protocols of analgesia in the immediate postoperative period. These findings support the need for careful postsurgical pain management in patients with pain identified as an indication for hysterectomy, independent of the extent of the surgery or underlying diagnosis.

## 1. Introduction

Endometriosis, defined as the presence of endometrial glands outside the uterine cavity, affects 10–15% of women of reproductive age [1]. Chronic pelvic pain (CPP) and painful menstruation (dysmenorrhea) are the main symptoms; secondary symptoms reported are painful intercourse (dyspareunia), painful defecation (dyschezia), painful urination (dysuria), and infertility [2]. CPP is defined as pain that occurs below the umbilicus and persists for at least 6 months [3,4]. In addition to endometriosis, many other gynaecological and non-gynaecological causes of CPP have been described, including adhesions, pelvic inflammatory disease, irritable bowel syndrome, interstitial cystitis, etc. [5]. The prevalence of endometriosis in women with CPP has been a topic of discussion; it varies between 25% and 70% [6,7]. The simultaneous co-existence of two or more associated causes is not rare [8].

Hysterectomy is the most common surgical procedure performed in non-pregnant women [9]; approximately, 12 to 17% of such surgeries are carried out due to CPP [10]; Acute postoperative pain is common. Nearly 20% of patients experience severe pain in the first 24 h after surgery [11]. Factors predictive of severe acute postoperative pain include younger age, female gender, increased body mass index, preoperative pain, preoperative use of opioids, and general anaesthesia [12]. Cognitive factors and affective responses, such as pain catastrophising, and psychological factors such as depression have also been proven to influence the course of acute pain after hysterectomy [13]. Multiple guidelines exist regarding postoperative analgesia, yet most are not specialty specific [14]; it is known, meanwhile, that gynaecologic surgeons tend to prescribe 2 to 4 times more opioids than the patients need after a laparoscopic hysterectomy [15,16].

It has been shown that women with endometriosis experience a decreased pain threshold and therefore a higher pain perception after noxious stimuli [17]. However, studies of patients with CPP have shown similar findings [18,19]. Treatment of central sensitisation is challenging; in the past, the focus has been placed on centrally acting drugs (tricyclic compounds, serotonin–norepinephrine reuptake inhibitors, etc.) without significant success [20]. Understanding whether the increased pain sensitivity is linked to the endometrial tissue itself, or whether it is the outcome of the chronic pain, unrelated to the presence of endometriosis, is essential in the context of postoperative pain management: it could help optimise the use of painkillers after surgery and increase patients’ satisfaction.

The aim of this study was to determine if, and to what extent, endometriosis or pre-existing pain affects the course of acute postoperative pain (pain occurring up to 7 days after the operation) experienced by women undergoing a total laparoscopic hysterectomy.

## 2. Materials and Methods

This retrospective study included patients between 19 and 51 years of age with signed general consent who underwent a total laparoscopic hysterectomy between January 2013 and October 2017 due to a non-malignant cause in the Department of Gynaecology of the Bern University Hospital, Switzerland. Clinical data were obtained from the patients’ electronic medical records. Information regarding patients’ demographics and medical history were collected (Body Mass Index (BMI), number of previous abdominal surgeries, parity, use of hormones before the operation). Attention was paid to the indication for the surgery, focusing on the expression of pain before the operation. The patients were divided into 2 groups: for the first group, hysterectomy was indicated or partly indicated by pain, while the second group reported no pain symptoms before the operation. When one of the terms 1. chronic pelvic pain, 2. dysmenorrhea, 3. dyspareunia, 4. dysuria, or 5. dyschezia was found under preoperative diagnosis or clinical indication included in the operative report, the patient was assigned to the first group; otherwise, she was classified as non-pain patient and registered in the second group. Further indications for surgery that were recorded include abnormal vaginal bleeding, cervical dysplasia, growing fibroids, sex reassignment surgery (-female- to -male-), Lynch syndrome, Breast Cancer gene (BRCA) mutation, uterine prolapse, and uterine polyps. The following surgical parameters were evaluated: duration of the operation, estimated blood loss, uterine morcellation, and complications during or after the operation, using the Clavien–Dindo classification [21]. In some cases, laparoscopic hysterectomy was combined with other interventions (adnexal surgery, removal of endometriosis, adhesiolysis, cystoscopy, and appendectomy). All surgeries were minimally invasive; 14 hysterectomies were robotic assisted using the da Vinci surgical system. The presence of endometriosis was confirmed either through histopathology or as a macroscopic finding, described in the surgical report. To describe the extent of endometriosis, the revised American Society for Reproductive Medicine (rASRM) classification (for peritoneal endometriosis), and the ENZIAN classification (for deep infiltrating endometriosis) were applied [22]. Peritoneal endometriosis classified as rASRM III or IV was considered to be severe endometriosis. Adenomyosis, defined as endometrial glands found in the uterine myometrium has, according to most of the current literature, different pathogenesis to endometriosis. However, there is a strong association between them, including similar clinical manifestation, frequent co-prevalence, and partly common molecular origin [23]. Therefore, cases of histologically confirmed adenomyosis without other endometriosis were also included in the endometriosis group in this study. Other histological findings of the uterus reported were fibroids, dysplasia of the cervix, polyps, and placental remnants.

The primary outcome was postoperative pain during the first 3 days of the hospital stay (day of surgery as well as the first two postoperative days). Two different methods of pain evaluation were applied: the visual analogue scale (VAS) score, recorded at least 3 times a day, and the use of analgesics. For the latter, the pain point system scale (PPSS) was applied [24]. The average VAS scores, as well as the lowest and highest VAS scores for each day of the hospital stay, were registered and compared. PPSS is a proven instrument for quantifying the use of painkillers in the context of retrospective studies: one dose of paracetamol 600 mg equals 1 point; a dose of paracetamol 500 mg plus 10 mg codeine equals 3 points; each dose of a nonsteroidal anti-inflammatory drug (diclofenac 75 mg, parecoxib 40 mg, or lornoxicam 8 mg) equals 3 points; and each dose of an opioid (tramadol 50 mg or dextropropoxyphene hydrochloride 75 mg or pethidine hydrochloride 50 mg) equals 7 points. When other painkillers in each category were administered, they were converted using universally accepted conversion scales: opioid morphine milligram equivalent and the Appendix of Comparable Non-Steroidal Anti-Inflammatory Drugs (NSAID) Dose Levels [25]. Secondary outcomes were the duration of hospital stay, the number of visits by the attending gynaecologist that were recorded in the electronic files, and the use of epidural anaesthesia or patient-controlled analgesia (PCA).

Each patient was assigned to one of the following four categories based on the indication for the hysterectomy and the presence of endometriosis or not: (1) endometriosis with preoperative pain (endo+/pain+); (2) preoperative pain without endometriosis (endo−/pain+); (3) endometriosis without preoperative pain (endo+/pain−); (4) no endometriosis and no preoperative pain (endo−/pain−).

Ethics approval for this research was obtained from the Ethics Commission of the Canton of Bern, Switzerland (Reference Number: 2020-00937).

### Statistical Analysis

Statistical analysis was performed using IBM SPSS Statistics (version 25.0). For patient and clinical–pathological analyses, basic descriptive statistics were applied. To compare the characteristics within the groups, the chi-squared test and ANOVA were used. A non-parametric test was applied for variables not meeting the assumptions of the *t*-test equivalent. Due to the increased likelihood of type II errors, adjustments for multiple testing were avoided [26]. All tests were two sided, and *p* values were considered statistically significant when <0.05.

## 3. Results

After the inclusion criteria were applied, 214 women were included in the study. Patient characteristics are summarised in Table 1. The median age was 40.63 (25–51) years at the time of surgery. There were no significant differences among the groups in age, body mass index, and the number of prior pregnancies. Women with endometriosis had a higher number of prior abdominal surgeries for pelvic pain or endometriosis. In the study, 26.2% (*n* = 56) of the patients underwent a simple laparoscopic hysterectomy; in 46.7% (*n* = 100) of the women, additional adnexal surgery was performed. For the rest of the collective, further interventions were recorded.

The surgery-related data are summarised in Table 2. For 53.3% (*n* = 57) of the women in the endometriosis group, a peritoneal or deep infiltrating endometriosis was reported. The rest displayed isolated adenomyosis uteri. Endometriosis was histologically confirmed in 47 cases. Severe peritoneal endometriosis (defined as endometriosis rAFS III and IV) was observed in 14 women and deep infiltrating endometriosis in 27 patients.

### 3.1. Postoperative Pain

Table 3 provides a summary of the postoperative pain and the medication received on the first three days after laparoscopic hysterectomy. Independent *t*-tests for statistical significance between groups were conducted; patients with pain symptoms before the operation reported significantly higher VAS scores (minimum, mean, maximum) during the first day (Figure 1), while the consumption of painkillers was statistically similar. The maximum Visual Analog Scale (VAS) score during the second observation day (*p*-value = 0.014), the mean VAS score during the third day (*p*-value = 0.017), and the maximum VAS score during the third day (*p*-value = 0.028) were also higher for patients with preoperative pain (Table 4). There was no statistical correlation between the presence or the extent of endometriosis and postoperative pain (Table 5).

### 3.2. Secondary Pain Outcomes

#### 3.2.1. Patient Controlled Analgesia (PCA)

Endo+/pain+: 8 patients (14%) used PCA for an average of 2 days.Endo−/pain+: 2 patients (4%) used PCA for an average of 2 days.Endo+/pain−: 3 patients (7.5%) used PCA for an average of 2.7 days.Endo−/pain−: 4 patients (6%) used PCA for an average of 2.25 days.

#### 3.2.2. Frequency of Doctor’s Visits

The number of doctor’s bedside visits (as extracted from the electronic files) was statistically higher for patients reporting pain symptoms before the surgery (3.01 vs. 2.84 during the total duration of the hospital stay, *p*-value < 0.001). The presence of endometriosis did not play a significant role.

#### 3.2.3. Duration of the Hospital Stay

Patients with endometriosis or preoperative pain tended to have a shorter hospital stay, compared with the group endo−/pain− (3.67 days vs. 3.78 days, *p*-value < 0.001). No other difference was found between the groups.

## 4. Discussion

In line with the results found in most of the current literature, our study shows that women with pain before the operation experience increased postoperative pain after laparoscopic hysterectomy. Our findings did not appear to be related to the presence or severity of endometriosis [27,28,29,30,31]. Our results are in line with the growing evidence that patients with diverse pain syndromes face a cardinal problem of pain amplification. Compared with the pain-free group, these patients report enhanced pain intensity and display augmented neuronal activity in pain-related regions of the brain when exposed to stimuli that healthy individuals find innocuous [32,33]. In the end, even in the absence of true pain, the receptors would remain activated, leading to chronic pain syndrome.

Surprisingly, the increased VAS scores of the groups with pain before the operation did not generate an analogous use of analgesics, as the PPSS was similar between the groups. This finding is presumably associated with the standardised pain management protocols applied nowadays. Most patients who undergo surgical procedures experience acute postoperative pain, but evidence suggests that less than half report adequate postoperative pain relief. Individual postoperative pain management is known to still need improvement [34] since insufficient pain management has negative psychological consequences for the patients affected and leads to increased morbidity and increased health care costs [35]. The latter is reflected in our study by the frequency of the doctors’ bedside visits, which was higher for the group of patients with CPP before the operation.

In our analysis, nearly 45% of women undergoing hysterectomy for benign indications reported pain before the operation, a prevalence higher than the one previously reported of 10–32% [10,36,37]. Uccella et al. found that the presence of endometriosis is associated with longer surgical time and increased morbidity related to total laparoscopic hysterectomy [38]; other studies support this statement [39]. In this study, no difference was found in the surgical time, blood loss, and the Clavien–Dindo complications between the groups. Moreover, previous reviews showed that women undergoing laparoscopic hysterectomy tended to be of lower parity and lower BMI when endometriosis or adenomyosis uteri was present [6]. There was no difference between parity and BMI in our study groups.

What makes this study unique is the use of multiple standardised parameters to assess postoperative pain—namely, the VAS score, the PPSS, and secondary variables. Further strengths of this study are the precise patient selection and the division into subgroups based on the visualisation of the endometriosis by the surgeon or/and the histopathological reports.

However, due to its retrospective nature, this study has some important limitations. Information about the presence of pain before the operation was extracted from the hospital’s electronic database. Although a VAS questionnaire (to assess the degree of dysmenorrhea, abdominal pain, dyspareunia, dysuria, and dyschezia during the previous four weeks) was handed to all patients before the surgery, as part of other prospective studies, only 35% of our study population answered the questionnaire. Furthermore, different high-volume endoscopic surgeons performed most laparoscopic hysterectomies included in this study, and thus, it is possible that our findings may not be directly applicable to other settings. Additionally, as acute postoperative pain is defined as the pain occurring during the first 7 days after the surgery, assessing only the first 3 days can lead to bias. Finally, although the groups were relatively homologous, they included predominantly Caucasian women living in a specific region of central Europe.

## 5. Conclusions

Women with chronic pelvic pain demonstrate an amplified pain perception during the first days after laparoscopic hysterectomy; postsurgical pain does not correlate with the presence or extent of endometriosis. However, the increased pain does not translate into a corresponding administration of painkillers, possibly due to the standardised protocols of analgesia in the immediate postoperative period. In the era of personalised medicine, postoperative pain management should focus on the surgical approach and individual patient characteristics, such as pain before operation. Optimising the management of acute postoperative pain is crucial, as it is one of the main risk factors for developing chronic postoperative pain [27,28,29].

## Figures and Tables

**Figure 1 jcm-10-04727-f001:**
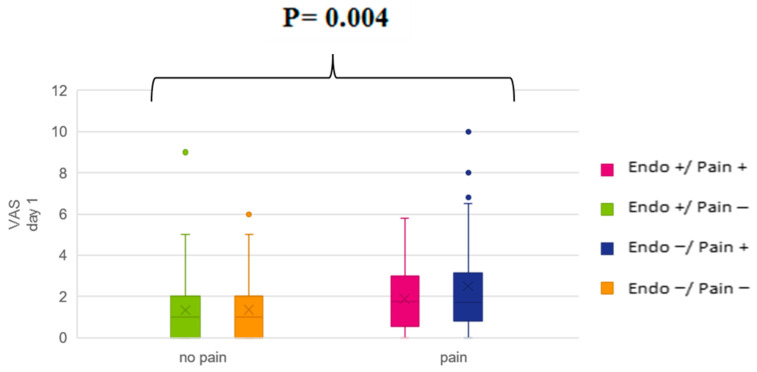
Comparison of the mean postoperative VAS scores during the day of the operation between patients with and without pain before the surgery (*p*-value = 0.004). Similar results apply for the minimum (*p*-value = 0.01) and maximum (*p*-value = 0.006) VAS scores.

**Table 1 jcm-10-04727-t001:** Patient characteristics.

	Endo+/Pain+ (*n* = 57)	Endo+/Pain− (*n* = 50)	Endo−/Pain+ (*n* = 40)	Endo−/Pain− (*n* = 67)	Total (*n* = 214)	*p*-Value
Age, years, mean	40.63 (SD = 5.42)	43.00 (SD = 5.19)	42.38 (SD = 5.46)	42.75 (SD = 7.19)	42.17 (SD = 6.02)	0.15
BMI, kg/m^2^, mean	26.54 (SD = 5.45)	27.14 (SD = 6.98)	27.01 (SD = 5.55)	27.26 (SD = 7.16)	26.99 (SD = 6.36)	0.94
Parity, *n*, mean	1.18 (SD = 1.07)	1.18 (SD = 1.40)	1.21 (SD = 1.23)	1.56 (SD = 1.42)	1.30 (SD = 1.30)	0.29
Previous abdominal surgeries due to pain or endometriosis, *n*, mean	0.79 (SD = 0.88)	0.36 (SD = 0.94)	0.18 (SD = 0.50)	0	0.33 (SD = 0.74)	<0.05
Number of women with previous hormonal treatment, *n* (%)	34 (59.6%)	17 (34%)	11 (27.5%)	20 (29.9%)	82 (38.3%)	<0.05

Numerical variables were assessed for significance using analysis of variance (ANOVA); categorical values were assessed using the chi-squared test. BMI: Body Mass Index.

**Table 2 jcm-10-04727-t002:** Surgical outcomes.

	Endo+/Pain+ (*n* = 57)	Endo+/Pain− (*n* = 50)	Endo−/Pain+ (*n* = 40)	Endo−/Pain− (*n* = 67)	Total (*n* = 214)	*p*-Value
Duration, min, mean	140.67 (SD = 64.55)	137.06 (SD = 53.97)	122.63 (SD = 40.61)	131.96 (SD = 55.41)	133.72 (SD = 55.40)	0.43
Blood loss, mL, mean	128.07 (SD = 95.91)	144.00 (SD = 139.84)	130.00 (SD = 122.89)	149.25 (SD = 147.58)	138.79 (SD = 128.51)	0.78
Number of Morcellations, *n* (%)	5 (8.8%)	16 (32%)	9 (22.5%)	16 (23.9%)	46 (21.5%)	0.03
Complications Clavien I–II, *n* (%)	1 (1.8%)	6 (12%)	6 (15%)	4 (6%)	17 (7.9%)	
Complications Clavien III–IV, *n* (%)	2 (3.5%)	2 (4%)	3 (7.5%)	4 (6%)	11 (5.1%)	0.21 *
Uterus weight, gr, mean	154.37 (SD = 118.94)	275.70 (SD = 231.15)	291.94 (SD = 231.36)	333.44 (SD = 376.37)	264.58 (SD = 274.20)	0.005

* The *p*-value refers to the total number of surgical complications; numerical variables were assessed for significance using analysis of variance (ANOVA); categorical values were assessed using the chi-squared test.

**Table 3 jcm-10-04727-t003:** Postoperative pain using the VAS score and the PPSS.

	Endo+/Pain+ (*n* = 57)	Endo+/Pain− (*n* = 50)	Endo−/Pain+ (*n* = 40)	Endo−/Pain− (*n* = 67)	Total (*n* = 214)	*p*-Value
VAS day 1, mean	1.87 (SD = 1.60)	1.34 (SD = 1.69)	2.50 (SD = 2.63)	1.36 (SD = 1.33)	1.72 (SD = 1.82)	0.014
VAS day 1, min	1.23 (SD = 1.64)	1.10 (SD = 1.64)	2.15 (SD = 2.65)	0.78 (SD = 1.33)	1.24 (SD = 1.78)	0.004
VAS day 1, max	2.62 (SD = 2.30)	1.71 (SD = 1.89)	2.94 (SD = 2.77)	1.95 (SD = 1.79)	2.27 (SD = 2.20)	0.039
PPSS day 1, mean	18.0 (SD = 9.2)	13.5 (SD = 9.9)	15.6 (SD = 8.2)	15.8 (SD = 9.1)	15.8 (SD = 9.3)	0.103
VAS day 2, mean	1.73 (SD = 1.31)	1.46 (SD = 1.47)	1.90 (SD = 1.34)	1.48 (SD = 1.32)	1.62 (SD = 1.36)	0.328
VAS day 2, min	0.69 (SD = 1.07)	0.77 (SD = 1.39)	0.73 (SD = 1.04)	0.64 (SD = 1.15)	0.70 (SD = 1.16)	0.944
VAS day 2, max	3.29 (SD = 2.48)	2.38 (SD = 2.13)	3.38 (SD = 2.02)	2.69 (SD = 2.23)	2.90 (SD = 2.26)	0.089
PPSS day 2, mean	16.9 (SD = 7.1)	16.0 (SD = 10.9)	15.4 (SD = 7.6)	15.7 (SD = 6.9)	16.0 (SD = 8.2)	0.799
VAS day 3, mean	1.81 (SD = 1.64)	1.39 (SD = 1.66)	1.94 (SD = 1.87)	1.14 (SD = 1.41)	1.53 (SD = 1.64)	0.096
VAS day 3, min	1.30 (SD = 1.42)	1.11 (SD = 1.56)	1.48 (SD = 1.77)	0.86 (SD = 1.06)	1.15 (SD = 1.43)	0.237
VAS day 3, max	2.30 (SD = 2.14)	1.76 (SD = 2.03)	2.29 (SD = 2.48)	1.43 (SD = 1.80)	1.90 (SD = 2.10)	0.148
PPSS day 3, mean	10.2 (SD = 8.5)	8.5 (SD = 6.4)	8.8 (SD = 6.1)	8.1 (SD = 7.1)	8.9 (SD = 7.2)	0.427

All variables were assessed for significance using analysis of variance (ANOVA). VAS: Visual Analog Scale; PPSS: pain point system scale.

**Table 4 jcm-10-04727-t004:** Comparison of postoperative pain between groups with and without pain before surgery.

	Preoperative Pain (*n* = 97)	No Preoperative Pain (*n* = 117)	*p*-Value
VAS day 1, mean	2.12 (SD = 2.08)	1.35 (SD = 1.49)	0.004
VAS day 1, min	1.59 (SD = 2.01)	0.92 (SD = 1.47)	0.010
VAS day 1, max	2.74 (SD = 2.49)	1.85 (SD = 1.83)	0.006
PPSS day 1, mean	17.0 (SD = 8.8)	14.8 (SD = 9.5)	0.088
VAS day 2, mean	1.80 (SD = 1.32)	1.47 (SD = 1.38)	0.079
VAS day 2, min	0.70 (SD = 1.05)	0.70 (SD = 1.25)	0.958
VAS day 2, max	3.33 (SD = 2.28)	2.55 (SD = 2.18)	0.014
PPSS day 2, mean	16.3 (SD = 7.3)	15.8 (SD = 8.8)	0.660
VAS day 3, mean	1.86 (SD = 1.73)	1.25 (SD = 1.52)	0.017
VAS day 3, min	1.38 (SD = 1.57)	0.97 (SD = 1.29)	0.068
VAS day 3, max	2.30 (SD = 2.27)	1.57 (SD = 1.90)	0.028
PPSS day 3, mean	9.7 (SD = 7.6)	8.3 (SD = 6.8)	0.169

**Table 5 jcm-10-04727-t005:** Comparison of postoperative pain between groups with and without endometriosis.

	Endometriosis (*n* = 107)	No Endometriosis (*n* = 107)	*p*-Value
VAS day 1, mean	1.80 (SD = 2.00)	1.64 (SD = 1.66)	0.575
VAS day 1, min	1.30 (SD = 2.04)	1.17 (SD = 1.49)	0.619
VAS day 1, max	2.32 (SD = 2.25)	2.22 (SD = 2.17)	0.743
PPSS day 1, mean	15.7 (SD = 8.8)	15.9 (SD = 9.8)	0.891
VAS day 2, mean	1.64 (SD = 1.34)	1.60 (SD = 1.39)	0.816
VAS day 2, min	0.67 (SD = 1.10)	0.73 (SD = 1.23)	0.732
VAS day 2, max	2.95 (SD = 2.17)	2.85 (SD = 2.35)	0.745
PPSS day 2, mean	15.6 (SD = 7.2)	16.5 (SD = 9.1)	0.425
VAS day 3, mean	1.44 (SD = 1.64)	1.61 (SD = 1.65)	0.514
VAS day 3, min	1.10 (SD = 1.39)	1.21 (SD = 1.48)	0.619
VAS day 3, max	1.76 (SD = 2.11)	2.05 (SD = 2.10)	0.375
PPSS day 3, mean	8.4 (SD = 6.7)	9.4 (SD = 7.6)	0.305

## Data Availability

All data involved in this study will be made available by the corresponding author upon request.

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
