# Peer review of "Postoperative Pain Is Driven by Preoperative Pain, Not by Endometriosis"

_jcm, 2021, doi:10.3390/jcm10204727_

Round 1

Reviewer 1 Report

Thank you for the opportunity to review this manuscript entitled “Postoperative pain is driven by preoperative pain, not by endometriosis.” The authors characterized 214 women who underwent hysterectomy by their surgical indication (endometriosis/no endometriosis) and preoperative pain status (pain/no pain). Women with preoperative pain, regardless of indication for surgery had worse postoperative pain, but no perceptible increase in the use of analgesics. While the study contributes to the existing literature on preoperative pain as a salient risk factor for postoperative outcomes, substantial issues related to characterization of subgroups and statistical analyses must be addressed.

Major Issues:

  1. Unclear characterization of subgroups:
    1. It is really unclear how preoperative pain was determined (“expression of pain” (p2. Line 64) among the subgroups. As it reads currently, only those women with endometriosis were characterized according to chronic pelvic pain, but whether this entailed a documented diagnosis of chronic pelvic pain is not clear. Pain-free is described as ‘reported no pain symptoms before the operation” (p.2, line 67-68) – it is not clear when and how women were reporting or not reporting pain.
  • If possible, a detailed characterization of preoperative pain would contribute far more to the field and allow for stronger speculation on mechanism, etc. If this information is available to the authors, I encourage them to include that in their paper.
    1. Endometriosis is also loosely characterized, including a separate condition of adenomyosis, without any explanation of why this was done.
  1. Several analyses were conducted with no adjustments made for multiple comparisons. ANOVAs are not followed by pairwise contrasts so the reader cannot see between which subgroups the difference lies. Suggest that authors consult with statistician as to how best to answer their research question and whether they have the statistical power to do so.
  2. 6, lines 198-199 – the sample is not really homogenous at all (see 1b above and diversity of conditions included in non-endometriosis subgroup) and the sample is not very large (at 40-67 per subgroup with an immense number of statistical tests performed).

Minor (easily addressable) issues:

  1. Abstract states outcomes were measured over first two postoperative days, introduction states over 7 days, methods states over 3 days – should be clear and consistent.
  2. Introduction states that “understanding whether the increased pain sensitivity is linked to the endometrial tissue itself or whether it is the outcome of the chronic pain, unrelated to the presence of endometriosis, is essential in the context of postoperative pain management.” The rationale, as presented, is not clear.
  3. The introduction is very brief and does not include a sufficient overview of literature on preoperative pain as a predictor of postoperative outcomes (in general and specific to hysterectomy) – e.g., PMID: 20090433, PMID: 28800726, PMID: 34121077, PMID: 30837064)
  4. Pain sensitivity is used interchangeably with report of pain intensity. These are not the same constructs.
  5. P-values in text do not have analogous information in the tables (i.e., when subgroups with preop pain were combined).
  6. Section 3.2.1 on PCA is very unusually reported.

Author Response

Dear reviewer,  

Thank you for the valuable inputs to our work. We feel we have addressed conscientiously all the questions and critiques of all reviewers and have integrated many important points and perspectives in response to the input received. On this basis, we are convinced that the paper has been improved greatly. We trust that you will find this revised version ready for publication. 

We have answered your question below and introduced the necessary changes in the manuscript  in consequence.

Major Issues

1.Unclear characterization of subgroups: It is really unclear how preoperative pain was determined (“expression of pain” (p2. Line 64) among the subgroups. As it reads currently, only those women with endometriosis were characterized according to chronic pelvic pain, but whether this entailed a documented diagnosis of chronic pelvic pain is not clear. Pain-free is described as ‘reported no pain symptoms before the operation” (p.2, line 67-68) – it is not clear when and how women were reporting or not reporting pain.

If possible, a detailed characterization of preoperative pain would contribute far more to the field and allow for stronger speculation on mechanism, etc. If this information is available to the authors, I encourage them to include that in their paper.

Thank you for point out the unclear characterization of preoperative pain. The classification of the patients was based on the operative reports. We changed the manuscript accordingly (p.2 lines 70 – 74). A more detailed description of the preoperative pain is unfortunately not possible, due to the retrospective nature of the study.   

2. Endometriosis is also loosely characterized, including a separate condition of adenomyosis, without any explanation of why this was done. 

We agree; even though the “from outside to inside invasion” theory, suggesting the migration of ectopic endometrial cells from posterior endometriosis nodules into the myometrium, is not completely rejected, according to newest papers endometriosis and adenomyosis are considered to be different conditions with different pathogenesis  However there is strong association between them. Both are estrogen-dependent, they display similar symptomologies (dysmenorrhea recorded in up to 79% of patients with adenomyosis) while Leyendecker et al. (PMID 25241270) have shown a very high prevalence of endometriosis in patients with adenomyosis and the opposite. Furthermore, many studies display a correlation between deep infiltrating endometriosis and focal adenomyosis of the outer myometrium. We have added a statement, explaining the inclusion of adenomyosis in the endometriosis group (p.2 lines 88 – 92).

3. Several analyses were conducted with no adjustments made for multiple comparisons. ANOVAs are not followed by pairwise contrasts so the reader cannot see between which subgroups the difference lies. Suggest that authors consult with statistician as to how best to answer their research question and whether they have the statistical power to do so. 

Adjustments for multiple comparisons in order to avoid type I errors are, according to literature, controversial and can lead to an increase of type II errors (PMID: 9553006). The p-values in the tables refer to the differences among all the groups. We have conducted several pairwise comparisons using independent t-tests and added a statement to clarify this to our readers (p. 3.1 lines 167-168). Due to the large scale of our analyses we considered reporting only the statistically relevant results which are included in the text (for example p.3.1. lines 170 – 172). An external medical statistician was consulted before submitting the revised manuscript. 

4. lines 198-199 – the sample is not really homogenous at all (see 1b above and diversity of conditions included in non-endometriosis subgroup) and the sample is not very large (at 40-67 per subgroup with an immense number of statistical tests performed). 

 This statement has been removed. 

Minor Issues

1.Abstract states outcomes were measured over first two postoperative days, introduction states over 7 days, methods states over 3 days – should be clear and consistent.

We measured the postoperative pain during the day of the surgery as well as during the following two days. We have added a clearer definition of the time interval in the abstract (line 19) and introduction (p. 1 line 97). The 7 days - interval refers to the definition of the acute postoperative pain; this point is highlighted in the manuscript (p. 4 lines 247 -249 ).

2.Introduction states that “understanding whether the increased pain sensitivity is linked to the endometrial tissue itself or whether it is the outcome of the chronic pain, unrelated to the presence of endometriosis, is essential in the context of postoperative pain management.” The rationale, as presented, is not clear 

A more precise explanation has been added (p.1, lines 53-55) 

3.The introduction is very brief and does not include a sufficient overview of literature on preoperative pain as a predictor of postoperative outcomes (in general and specific to hysterectomy) – e.g., PMID: 20090433, PMID: 28800726, PMID: 34121077, PMID: 30837064) 

Thank you for your comment. Two of the recommended papers were already included in our reference section ( PMID: 28800726, PMID: 30837064 ). PMID: 20090433 and PMID: 34121077 are completely in line with our general context and are now cited.

4.Pain sensitivity is used interchangeably with report of pain intensity. These are not the same constructs. 

Thank you for pointing this out; the term `pain sensitivity` has been replaced in line 207. We still believe it is in line with the context in line 48.

5.P-values in text do not have analogous information in the tables (i.e., when subgroups with preop pain were combined). 

As mentioned above (major issues, comment 3), we compared our groups separatelyWe believe that by reporting only the significant statistical results in the text we would avoid unnecessary repetitions and therefore make the manuscript easier to read (given that the mean values of each variable/group are included in the tables). 

6. Section 3.2.1 on PCA is very unusually reported 

Section 3.2.1 aims to provide all the information available on an analgesic instrument, which undoubtedly affects the course of postoperative pain and therefore the results of our study.  Quantifying patient controlled analgesia in the context of retrospective studies is challenging, as the number of boluses that each individual self-administers is usually not recorded; for that reason prospective studies usually set a limit of allowed boluses per hour/day (for example PMID:17903498).

Reviewer 2 Report

Kanellos and coworkers   submitted an inetresting manuscript stating that women with chronic pelvic pain demonstrate an amplified pain perception during  the first days after laparoscopic hysterectomy; post-surgical pain does not correlate with the presence or extent of endometriosis.

A total of 214  patients were investigated in their  retrospective study

Authors demonstarted that  the increased pain does not translate  into a corresponding administration of painkillers, possibly due to the standardized protocols of analgesia in the immediate postoperative period. The difference of the mean postoperative VAS - score during the day of the operation between patients with and without pain before the surgery was statistically significant (p-value = .004).

 I would like to ask the authors to comment the role of central sensitisation in pre- and postoperative pain management  among patients with chronic pelvic pain and endometriosis respectively.

The manuscript meets the standards of JCM and after minor english language and style corrections

Author Response

Dear reviewer,  

Thank you for the valuable inputs to our work. We feel we have addressed conscientiously all he questions and critiques of all reviewers and have integrated many important points and perspectives in response to the input receivedOn this basis, we are convinced that the paper has been improved greatly. We trust that you will find this revised version ready for publication. 

We have answered your question below and introduced the necessary changes in the manuscript  in consequence.

 I would like to ask the authors to comment the role of central sensitisation in pre- and postoperative pain management  among patients with chronic pelvic pain and endometriosis respectively.

 Treatment of central sensitization is challenging; in the past focus has been placed on centrally acting drugs (tricyclic compounds, serotonin-norepinephrine reuptake inhibitors and others) without significant success. It is nowadays widely accepted that psychological factors interact with structural alterations in pain related brain regions. This could explain why a diagnostic laparoscopy or oral placebo medication have been found to alleviate pain in 50% of patients with endometriosis (PMID: 7926075, PMID: 7926076) . Thus, pain neuroscience education, cognition-targeted exercise therapy, sleep and stress management and dietary interventions have been reported to aid the pain management in patients with chronic pelvic pain. In the case of endometriosis the role of physical therapy and dry needling has been highlighted.

Round 2

Reviewer 1 Report

Thank you for your responses and edits. I have a few remaining concerns that were perhaps unclear or not addressed.

1. The Introduction is still quite brief and 'thin' - I do not see additional background on what is already known and what gap you are filling, as was suggested in the initial review. 

2. I would include in your Methods (Statistical Analysis section) that you did not adjust for multiple comparisons (and your reasons for not doing so). 

2. It would be clearer for you to include the findings from your post-hoc pairwise comparisons in the tables so the reader can see which groups differ. 

3. I do still find it confusing to report statistics on combined groups in text and separated groups in tables. I will defer to your and the editor's judgment here, but it seems that you could add columns with combined groups to the existing tables so that the reader has all the information in one place.

4. I was perhaps unclear about PCA reporting. I only meant that the format of the text was unusual (bullet points vs complete sentences).

Author Response

Dear reviewer, We are grateful for your insightful comments. We have been able to incorporate changes to reflect most of the suggestions provided. We hope the manuscript after careful revisions meets your high standards. Here is a point-by-point response to the your comments and concerns. 1.The Introduction is still quite brief and 'thin' - I do not see additional background on what is already known and what gap you are filling, as was suggested in the initial review. The introduction has been enriched with information regarding acute postoperative pain (in general and specifically after hysterectomy) and central sensitization. Relevant literature has been cited. ( Part 1, lines 44-51, 57-60 ). 2. I would include in your Methods (Statistical Analysis section) that you did not adjust for multiple comparisons (and your reasons for not doing so). A statement regarding our decision not to perform adjustments for multiple comparisons has been added (Part 2, Lines 133-134). 3-4. It would be clearer for you to include the findings from your post-hoc pairwise comparisons in the tables so the reader can see which groups differ I do still find it confusing to report statistics on combined groups in text and separated groups in tables. I will defer to your and the editor's judgment here, but it seems that you could add columns with combined groups to the existing tables so that the reader has all the information in one place. Following your recommendation we have added Table 4 and Table 5 where we present the mean values in regard to postoperative pain of the combined groups ( pain +/- and endometriosis +/-). We find that now the reader has a better overview on the statistically important results, relevant to the purpose of our study. 5.I was perhaps unclear about PCA reporting. I only meant that the format of the text was unusual (bullet points vs complete sentences). We apologize for the misinterpretation of your comment. In this short section regarding PCA, we report a lot of numerical data. Therefore, in our opinion, using complete sentences would be fatiguing for the reader.
